# Placental Glycoredox Dysregulation Associated with Disease Progression in an Animal Model of Superimposed Preeclampsia

**DOI:** 10.3390/cells10040800

**Published:** 2021-04-03

**Authors:** Sandra M. Blois, Paula D. Prince, Sophia Borowski, Monica Galleano, Gabriela Barrientos

**Affiliations:** 1Department of Obstetrics and Fetal Medicine, University Medical Center Hamburg-Eppendorf, 20246 Hamburg, Germany; sophia.borowski@charite.de; 2Fisicoquímica, Facultad de Farmacia y Bioquímica, Universidad de Buenos Aires, Ciudad Autónoma de Buenos Aires C1113AAD, Argentina; prince_pdenise@hotmail.com (P.D.P.); mgallean@ffyb.uba.ar (M.G.); 3Instituto de Bioquímica y Medicina Molecular (IBIMOL), Universidad de Buenos Aires—Consejo Nacional de Investigaciones Científicas y Técnicas, Ciudad Autónoma de Buenos Aires C1113AAD, Argentina; 4Experimental and Clinical Research Center, a Cooperation between the Max Delbrück Center for Molecular Medicine in the Helmholtz Association, and the Charité—Universitätsmedizin Berlin, 13125 Berlin, Germany; 5Laboratorio de Medicina Experimental, Hospital Alemán—Consejo Nacional de Investigaciones Científicas y Técnicas, Ciudad Autónoma de Buenos Aires C1118AAT, Argentina

**Keywords:** chronic hypertension, placenta, preeclampsia, oxidative stress, glycosylation

## Abstract

Pregnancies carried by women with chronic hypertension are at increased risk of superimposed preeclampsia, but the placental pathways involved in disease progression remain poorly understood. In this study, we used the stroke-prone spontaneously hypertensive rat (SHRSP) model to investigate the placental mechanisms promoting superimposed preeclampsia, with focus on cellular stress and its influence on galectin–glycan circuits. Our analysis revealed that SHRSP placentas are characterized by a sustained activation of the cellular stress response, displaying significantly increased levels of markers of lipid peroxidation (i.e., thiobarbituric acid reactive substances (TBARS)) and protein nitration and defective antioxidant enzyme expression as early as gestation day 14 (which marks disease onset). Further, lectin profiling showed that such redox imbalance was associated with marked alterations of the placental glycocode, including a prominent decrease of core 1 O-glycan expression in trophoblasts and increased decidual levels of sialylation in SHRSP placentas. We also observed significant changes in the expression of galectins 1, 3 and 9 with pregnancy progression, highlighting the important role of the galectin signature as dynamic interpreters of placental microenvironmental challenges. Collectively, our findings uncover a new role for the glycoredox balance in the pathogenesis of superimposed preeclampsia representing a promising target for interventions in hypertensive disorders of pregnancy.

## 1. Introduction

With a global prevalence of 1 to 5%, chronic hypertension during pregnancy contributes to a significant healthcare economic burden associated with poor maternal and perinatal outcomes. Adverse outcomes are mostly due to progression to superimposed preeclampsia (SPE) which occurs in 25–40% of pregnancies affected by chronic hypertension [1]. Accurate diagnosis of SPE is still challenging, but the condition is presumed upon worsening of the hypertensive state associated with signs of end-organ damage including proteinuria and renal insufficiency, liver involvement, neurological disturbances or uteroplacental dysfunction and fetal growth restriction (FGR), presenting usually as early onset disease (i.e., before the 34th week) [1,2]. As obesity rates and maternal childbearing age continue to rise, chronic hypertension with SPE is expected to remain a major cause of morbidity and mortality among women and their offspring [3,4].

The core event in preeclampsia pathogenesis is placental malperfusion [5], which originates from early defects in the physiological conversion of maternal spiral arteries mediated by invasive trophoblast cells. At the second stage, ischemia–reperfusion damage of the placenta with oxidative/endoplasmic reticulum stress enhances the release of soluble mediators that trigger the systemic inflammatory cascade and endothelial dysfunction ultimately responsible for the maternal syndrome [6,7]. While the specific mechanisms promoting SPE are still largely unknown, evidence from clinical studies [8,9,10] and, more recently, animal models of chronic hypertension [11,12,13] point out to the involvement of placental cellular and molecular pathways largely overlapping those described for the de novo syndrome.

Dysregulated redox responses are involved in the initiation and progression of many inflammatory diseases including preeclampsia [14]. Recent findings indicate that changes in the production of reactive oxygen species (ROS) due to the activation of cellular stress response (CSR) pathways contribute to the regulation of glycan functions [15]. Glycans are crucial determinants of cell function and the glycocode expressed in a particular compartment (e.g., placenta) is a dynamic reflection of its developmental and pathophysiological status [16]. In the placenta, glycans modify proteins required for trophoblast function and altered glycosylation has been associated with pathological conditions including different types of hypertensive disorders of pregnancy and fetal growth restriction [17,18,19]. As paramount “interpreters” of the glycocode, members of the galectin family of β-galactoside binding proteins have attracted great attention due to their unique ability to modulate developmental processes supporting pregnancy maintenance and their potential use as disease biomarkers (reviewed in [20]). Of particular interest in the context of preeclampsia is the role played by galectin–glycan circuits in the modulation of pro-survival and pro-apoptotic pathways [21,22], a function essential for cellular homeostasis under microenvironmental insults ranging from metabolic imbalance and local hypoxia to oxidative stress. Indeed, several of the galectins most abundantly expressed at the maternal fetal interface (i.e., Gal-1, Gal-3, Gal-9 and Gal-13) are considered markers of the CSR, acting as “alarmin-like” molecules to signal tissue damage and promote an effector response from immune cells [23,24,25,26]. In this context, dynamic changes in the galectin–glycan network associated with an overall CSR and redox imbalance upon placental malperfusion could play an important role in the pathogenesis of preeclampsia.

In this study, we used the stroke-prone spontaneously hypertensive rat (SHRSP) model for essential hypertension that displays an SPE phenotype during pregnancy [11]. SHRSP pregnancies are also characterized by early defects in trophoblast invasion leading in turn to placental dysfunction, asymmetric FGR and a maternal syndrome with increased hypertension and proteinuria towards term [11]. We investigated the involvement of placental redox imbalance and the CSR in SPE pathogenesis and characterized the dynamic changes in the expression profile of glycans and stress-sensitive galectins in the placenta associated with disease progression to shed light on the mechanisms promoting SPE.

## 2. Materials and Methods

### 2.1. Animals and Experimental Design

This follow-up study was conducted using samples collected for our previous work on the characterization of SHRSP pregnancies as a spontaneous model of chronic hypertension with SPE [11]. Timed pregnancies were established by cohabitation of nulliparous SHRSP and Wistar Kyoto (WKY, normotensive controls) females (8–12-week-old, 200–250 g body weight) with congenic males, denoting the morning of detection of a vaginal plug as gestation day (GD) 1. Maternal systolic blood pressure (SBP) profiles throughout pregnancy were established as described previously [11]. A total of 37 WKY and 44 SHRSP breeding pairs were used, so that *n* = 5–7 pregnant animals per GD were included in the study.

### 2.2. Tissue Collection

Rats were euthanized on GD14 or GD18 while under thiopental sodium anesthesia for collection of fetoplacental samples. The uterus was dissected exposing the fetoplacental units, and whole implantation sites were prepared by carefully separating each unit (i.e., the placenta with its associated mesometrial triangle and decidual tissue). Fetoplacental specimens were fixed in phosphate-buffered 10% formaldehyde (pH 7.2) and embedded in paraffin according to the standard methods for histological sectioning [27].

### 2.3. Determination of Thiobarbituric Acid Reactive Substances (TBARS)

The assay was based on a previously described method [28]. Briefly, the samples were mixed with 3% (*w*/*v*) sodium dodecyl sulfate (SDS), 0.1 N HCl, 10% (*w*/*v*) phosphotungstic acid and 0.7% (*w*/*v*) 2-thiobarbituric acid and heated for 45 min in boiling water; TBARS were extracted with 2 mL of n-butanol and detected fluorometrically (λex: 515 nm; λem: 555 nm). To prepare the standard of malondialdehyde, 1,1,3,3-tetramethoxypropane was used. The results were expressed as TBARS nmol (malondialdehyde equivalents)/mg protein.

### 2.4. Immunohistochemistry for Oxidative Stress Markers, Antioxidant Enzymes and Galectins

Paraffin-embedded implantation sites were sectioned (4-μm-thick) parallel to the mesometrial–fetal axis and inmunolabeled following a modified avidin–biotin–peroxidase complex technique using a Vectastain ABC kit (Universal Elite; Vector Laboratories, Burlingame, CA, USA). Following deparaffinization and rehydration, the sections were washed in PBS for 5 min. Quenching of endogenous peroxidase (PO) activity was achieved by incubating the sections for 30 min in 1% H_2_O_2_ in methanol. The sections were then washed in PBS (pH 7.2) for 20 min, followed by incubation with blocking serum for 20 min. The sections were then incubated with primary antibodies against nitrotyrosine (#MAB5404; Millipore, Burlington, MA, USA), Hsp27 (sc-1048), Hsp70 (sc-1060, both from Santa Cruz Biotechnology, Dallas, TX, USA), peroxiredoxin (ab15571) or thioredoxin (ab86255, both from Abcam, Cambridge, UK) overnight at 4 °C, rinsed in PBS and incubated with a biotinylated universal antibody for 30 min. The specimens were washed in PBS, incubated for 40 min with a Vectastain Elite ABC reagent (Vector Laboratories) and exposed for 5 min to 0.1% diaminobenzidine (Polysciences, Warrington, PA, USA) and 0.2% H_2_O_2_ in 50 mM Tris buffer, pH 8. After washing, the nuclei were counterstained with 0.1% Mayer’s hematoxylin, followed by a standard dehydration procedure and mounting in a DPX histology medium (Millipore-Sigma, St. Louis, MO, USA).

Immunostaining for galectin expression was achieved following our standard protocol [29]. Briefly, the sections were deparaffinized and rehydrated, washed in TBS and blocked for quenching endogenous peroxidase activity. After blocking with 2% normal serum for 20 min, primary antibodies against Gal-1 (sc-28248), Gal-3 (sc-20157) or Gal-9 (sc-19292, all from Santa Cruz Biotechnology) were incubated overnight at 4 °C. The slides were then incubated for 1 h at room temperature with a goat anti-rabbit PO-conjugated secondary antibody (1:200, cat. #111-035-003, Jackson ImmunoResearch, West Grove, PA, USA) for Gal-1 and Gal-3 determination or a donkey anti-goat PO-conjugated secondary antibody (1:200, cat. #705-035-147, Jackson ImmunoResearch) for Gal-9. The signal was detected using a liquid diaminobenzidine (DAB+) Substrate Chromogen System (cat. #K3467, DAKO, Glostruck, Denmark) at room temperature. The nuclei were counterstained with 0.1% Mayer’s hematoxylin followed by a standard dehydration procedure and mounting in a Vitro-Clud medium (R. Langenbrinck GmbH, Emmendingen, Germany).

### 2.5. Western Blots

Placental samples were homogenized in a lysis buffer (150 mM NaCl, 50 mM Trizma–HCl, 1% (*v*/*v*) NP-40, pH 8.0) in the presence of protease and phosphatase inhibitors and centrifuged at 600× g for 20 min. The samples containing the same protein concentration were added with a 2X solution of the sample buffer (62.5 mM Tris–HCl, pH 6.8, containing 2% (*w*/*v*) SDS, 25% (*w*/*v*) glycerol, 5% (*v*/*v*) β-mercaptoethanol and 0.01% (*w*/*v*) bromophenol blue) and heated at 95 °C for 2 min. Sample aliquots containing 30 μg of protein were separated by reducing 10% (*w*/*v*) polyacrylamide gel electrophoresis and electroblotted to polyvinylidene difluoride membranes. Colored molecular weight standards (GE Healthcare, Piscataway, NJ, USA) were run simultaneously. The membranes were blotted for 2 h in 5% (*w*/*v*) nonfat milk incubated overnight in the presence of the corresponding primary antibody: phospho-p47*^phox^* (Ser370; #PA5-38798, Invitrogen, Waltham, MA, USA), SOD1 (AB1237, Chemicon International, Temecula, CA, USA), p47*^phox^*, SOD2 and β-actin (#7660, #133134, #47778, Santa Cruz Biotechnology, CA, USA), all used in a 1:1000 dilution in PBS. After a subsequent incubation for 90 min at room temperature (RT) in the presence of the corresponding HRP-conjugated secondary antibody (mouse anti-rabbit IgG-HRP or goat anti-mouse IgG-HRP (#2357, #2005; Santa Cruz Biotechnology, 1:5000), the complexes were visualized by chemiluminescence. The films were scanned and a densitometric analysis was performed using ImageJ (National Institutes of Health, Bethesda, MD, USA). Band densities were normalized to the β-actin content.

### 2.6. Glycan Profiling by Lectin-Binding Analysis

Glycan profiling was performed as described previously [30]. O-glycan structures were determined using the *Helix pomatia* agglutinin (HPA; Tn antigen) and the *Arachis hypogaea* lectin (PNA; core 1). Binding of the *Lycopersicon esculentum* lectin (LEA) was used for the detection of polyLacNAc extensions (carried by core 2 O- and complex *N*-glycans). In addition, binding of the *Phaseolus vulgaris* lectin (PHA-L) was used for determination of β1,6-GlcNAc-branched complex *N*-glycans and sialyation was determined using the *Maackia amurensis* lectin (MAA) and the *Sambucus nigra* agglutinin (SNA-I), which recognize α2,3- and α2,6-linked sialic acid, respectively. Briefly, the slides were deparaffinized, rehydrated, washed in PBS and blocked with PBS, 5% BSA, 1% fish gelatin, for 20 min in a humid chamber at RT. Afterwards, the slides were blocked with a Carbo-Free Blocking Solution (SP-5040, Vector Laboratories) for 30 min at RT. Subsequently, the slides were incubated with biotinylated lectins HPA (20 ng/mL; BA-3601-1), PHA-L (20 ng/mL; BA-1801-2) or SNA-I (10 ng/mL; BA-6802-1, all from EY Laboratories) diluted in a Carbo-Free Blocking Solution overnight at 4 °C. Lectin-stained sections were then incubated with 2 μg/mL streptavidin–tetramethylrhodamine (S-870; Invitrogen) for 1 h at RT. Subsequently, the slides were incubated with fluorescein isothiocyanate (FITC)-labeled lectins PNA (20 ng/mL; F-2301-1), LEA (20 ng/mL; F-7001-1) or MAA (20 ng/mL; F-7801-2, all from EY Laboratories) diluted in a Carbo-Free Blocking Solution for 2 h at RT. The nuclei were counterstained with 4′,6-diamidino-2-phenylindole (DAPI) for 5 min at RT and the slides were mounted in Prolong Gold (P36930, Invitrogen). Stainings of whole implantation sites were digitally scanned with a high-resolution bright field and fluorescence slide scanner (Pannoramic MIDI BF/FL, 3DHISTECH Ltd., Budapest, Hungary), and staining was evaluated on virtual slides using Pannoramic Viewer 1.15.4 (3DHISTECH Ltd.) by two examiners blinded to the experimental group.

### 2.7. Morphometrical Analyses

Photodocumentation was performed using either the Nikon E400 (Nikon Instrument Group, Melville, NY, USA) or the Keyence BZ 9000 (Keyence Corporation, Itasca, IL, USA) microscope systems. All the measurements were made on one set of five serial parallel sections taken through the center of the placental site (as indicated by the presence of a maternal spiral artery in the decidual layer) and perpendicular to its flat (fetal) side. For all the markers, staining localization was assessed by two independent observers taking into account their distribution in the maternal (decidua, mesometrial triangle) and fetal (junctional zone, labyrinth) compartments of the placenta. Quantitative analyses were performed on a set of at least ten random images taken at ×200 magnification from each compartment (mesometrial triangle for maternal and labyrinth layer for fetal sources, respectively) and run using the color deconvolution plugin of the ImageJ software (National Institutes of Health, https://imagej.nih.gov/, accessed on 3 April 2021). All the results for histological staining were expressed as the percentage of area with positive immunostaining within each placental compartment.

### 2.8. Statistics

Data are expressed as the means ± SEM. Between-group differences were assessed by unpaired Student’s *t*-tests. Temporal changes in marker expression through gestation were analyzed using two-way ANOVA with factors strain (WKY vs. SHRSP) and GD and their interaction, followed by Bonferroni post-hoc tests. All the calculations were run using GraphPad Prism v 7.0 (GraphPad Software Inc., San Diego, CA, USA). A *p*-value of less than 0.05 was considered statistically significant.

## 3. Results

### 3.1. Onset of SPE in SHRSP Pregnancies Is Marked by an Enhanced Activation of the Placental CSR

Our analyses of SHRSP pregnancies focused on two specific gestational stages (Figure 1).

GD14 denotes the onset of SPE characterized by placentation and spiral artery remodeling defects without full development of maternal features and GD18 represents the established maternal syndrome with end-organ (i.e., renal) damage [11].

First, we evaluated the expression of markers for the CSR (Figure 2A) to investigate the involvement of placental stress and redox imbalance in SPE pathogenesis.

In both pregnancy models, progression of pregnancy from GD14 to GD18 was associated with a significant increment of placental levels of TBARS (Figure 2B), indicating increased lipid peroxidation. However, levels of TBARS on GD14 were 2-fold higher in SHRSP placentas compared to the controls (*p* < 0.05, Figure 2B) and remained increased as gestation progressed to GD18. Further, immunohistochemical analyses showed that SHRSP implantations displayed a significant increase of protein nitrotyrosine (pNT) residues on GD14 both in the maternal (mesometrial triangle, Figure 2C, upper panel) and placental (junctional zone and labyrinth, Figure 2C, bottom panel) compartments, indicating increased nitrosative stress. Notably, normal pregnancy progression was associated with a significant upregulation of pNT levels (*p* < 0.001, Figure 2C, bottom panel) in the placental labyrinth from GD14 to GD18. This upregulation was abrogated in SHRSP placentas, in which pNT immunolabeling remained at similar levels as gestation progressed from GD14 to GD18. The onset of SPE was also associated with augmented expression of the CSR chaperones Hsp27 and Hsp70 (Figure 2D) in SHRSP pregnancies, with a differential distribution in the maternal and placental compartments: while the mesometrial triangle displayed significantly increased Hsp27 (Figure 2D, left upper panel) but no differences in Hsp70 immunolabeling (Figure 2D, left bottom panel), both markers were significantly upregulated in the placental junctional zone and labyrinth compared to WKY. While in both models the expression of Hsp27 increased significantly from GD14 to GD18 (Figure 2D, left upper panel), levels of Hsp70 expression in the mesometrial triangle did not vary with pregnancy progression and were similar in both mating models (Figure 2D, left bottom panel). In the placental compartment, in contrast, SHRSP displayed reduced levels of Hsp70 expression compared to WKY implantations on GD18 (Figure 2D, right bottom panel). Furthermore, Western blot analyses revealed that normal pregnancy progression was associated with a significant upregulation of phosphorylated p47^phox^ (Figure 2E) from GD14 to GD18, indicating increased NADPH oxidase 2 (NOX2) activation. Of note, the normal upregulation of phospho-p47^phox^ from GD14 to GD18 was abrogated in SHRSP placentas, which displayed similarly high levels of expression on both days analyzed.

Next, we evaluated changes in the expression of antioxidant defense systems associated with progression of SPE in the SHRSP model. As shown in Figure 3A, placental expression of the protective ROS-detoxifying systems SOD1 (left panel), SOD2 (middle panel) and catalase (right panel) was upregulated from GD14 to GD18 in normally progressing WKY pregnancies. SHRSP implantations exhibited a similar response except for the expression of SOD2 (Figure 3A, middle panel), which showed no changes with pregnancy progression and was significantly decreased on GD18 compared to normal pregnancies. Additionally, SHRSP implantations presented significantly decreased immunolabeling for peroxiredoxin 1 on GD14 (PRX, Figure 3B), which was evident both in the mesometrial triangle and the junctional zone and labyrinth layer. Expression of thioredoxin (TRX, Figure 3C) was also diminished at the onset of SPE in the SHRSP model, but only in the maternal compartment. In the placental compartments, in contrast, SHRSP implantations displayed an upregulation of TRX expression with progression of pregnancy from GD14 to GD18 (Figure 3C, right panel).

### 3.2. Marked Changes in Placental Glycosylation Associated with the Onset of SPE in SHRSP Pregnancies

Next, we performed lectin-based stains to profile the placental expression of glycans in both models (Figure 4A). In normal pregnancies, we observed a significant downregulation of the Tn antigen (HPA, Figure 4B upper panels) and polyLacNAc extensions (LEA, bottom panels) in the decidua together with upregulated expression of the Thomsen-Friedenreich (TF) epitope (PNA, middle panels) in the mesometrial triangle with pregnancy progression. In the placenta, there were no changes in glycan expression from GD14 to GD18 except for a significant increase of TF expression in the labyrinth layer (Figure 4B, middle panels). In SHRSP, the onset of SPE on GD14 was marked by significantly reduced levels of TF expression in all maternal and placental compartments. Furthermore, as gestation progressed to GD18, there was a significant upregulation of TF levels in the placental giant cells and spongiotrophoblasts of the junctional zone (Figure 4B, middle panels). On GD18, SHRSP placentas displayed significantly decreased Tn antigen levels in the labyrinth (Figure 4B, upper panels), reduced TF expression in the mesometrial triangle and labyrinth (middle panels) and decreased levels of polyLacNAc-extended glycans in the placental giant cells and labyrinth (Figure 4B, bottom panels) compared to normotensive pregnancies.

As for complex *N*-glycans, PHA-L staining revealed a significant upregulation of expression levels in the mesometrial triangle and all placental compartments with progression of normal pregnancy (Figure 4C). SHRSP placentas showed similar kinetics of expression compared to WKY, except for significantly decreased levels of *N*-glycan expression in the mesometrial triangle (Figure 4C, upper panels), placental giant cells and junctional zone (bottom panels) at the onset of the SPE syndrome. On GD18, in contrast, SHRSP placentas displayed increased *N*-glycan levels in the giant cell layer compared to the normotensive controls (Figure 4C, bottom panels).

Our analysis further revealed that progression of normal pregnancy was linked with a significant upregulation of α2,3- (MAA, Figure 4D, upper panels) and α2,6-linked sialic acid (SNA-I, Figure 4D, bottom panels) content in all compartments of the WKY placenta from GD14 to GD18. Notably, despite similar kinetics of expression with pregnancy progression, SHRSP placentas displayed increased levels of α2,3-sialylation in the decidua and mesometrial triangle (Figure 4D, upper panels) and α2,6-sialylation in the mesometrial triangle (bottom panels) on GD14 compared to WKY. On GD18, SHRSP placentas exhibited increased α2,6-linked sialic acid content in the junctional zone compared to the controls (Figure 4D, bottom panels).

### 3.3. Development of SPE Induces Changes in the Expression Profile of Stress-Sensitive Galectins in the SHRSP Placenta

Our next aim was to define the galectin signature associated with progression of SPE in the SHRSP model (Figure 5A), evaluating changes in the expression profile of stress-sensitive galectins-1, -3 and -9 in the maternal and fetal compartments of the placenta at the onset of disease (GD14) and upon full establishment of the syndrome on GD18. Gal-1 was abundantly expressed in the GD14 placenta of both models, localizing to the cytoplasm and extracellular matrix of interstitial cells within the mesometrial triangle, giant cells of the junctional zone and the placental labyrinth. In the maternal compartment, normal pregnancy progression was associated with a significant upregulation of Gal-1 expression from GD14 to GD18 (Figure 5B, left panel). SHRSP pregnancies showed expression levels comparable to WKY on GD14 but the normal increase of Gal-1 associated with pregnancy progression was abrogated in this model, exhibiting significantly decreased levels compared to controls on GD18. In the SHRSP placenta, expression of Gal-1 remained unchanged with pregnancy progression (Figure 5B, right panel), resulting in significantly decreased levels of this galectin compared to controls on GD18.

Staining for Gal-3 was most prominent in the maternal compartment, but was also detected in moderate levels within the junctional zone and strongly expressed in the labyrinth layer of both mating models, showing mostly a cytoplasmic and extracellular matrix association. Maternal expression of Gal-3 was similar in both models with a significant downregulation as gestation progressed from GD14 to GD18 (Figure 5C, left panel). In the placenta, in contrast, normal pregnancy progression was associated with a marked upregulation of Gal-3 expression (Figure 5C, right panel). Of note, SHRSP placentas exhibited significantly increased expression of Gal-3 compared to WKY on GD14 but a significant downregulation of this galectin associated with pregnancy progression (Figure 5C, right panel), which resulted in decreased levels versus control placentas at GD18.

Expression of Gal-9 on GD14 showed a similar pattern in both models, being detected at moderate levels mainly with cytosolic localization to cells within the mesometrial triangle and the placental labyrinth. No differences in mesometrial triangle Gal-9 were observed as gestation progressed in any of the models, which showed similar expression levels of this galectin at both days analyzed (Figure 5D, left panel). In contrast to WKY, labyrinth expression levels of Gal-9 in the SHRSP model were not modified as gestation progressed to GD18 and were significantly lower than in WKY pregnancies at both days analyzed (Figure 5D, right panel).

## 4. Discussion

Impaired placental function is generally assumed as the major contributor to development of SPE, yet pathophysiological evidence in support of this notion is relatively scarce. In this study, we demonstrated that onset of SPE in SHRSP pregnancies is characterized by a persistent activation of the placental CSR linked with a prominent decrease of placental core 1 O-glycans. These changes were associated with increased sialylation in the maternal compartments, providing novel insights on dysregulated molecular pathways associated with placental insufficiency in pregnancies affected by chronic hypertension. We further observed marked changes of the placental galectin signature, highlighting the paramount role of these proteins as dynamic “sensors” of microenvironmental challenges affecting the placental glycoredox status.

Our analysis of the placental CSR first revealed significant temporal changes in markers of lipid peroxidation, nitrosative stress, heat shock protein expression and NOX2 activation in WKY placentas, indicating that activation of the CSR is a feature of normal pregnancy progression. Pregnancy per se imposes a physiological cellular stress, as normal placental development involves extensive cell division, high metabolic rates and temporal fluctuations in oxygen tension [31]. In the rat placenta, establishment of the hemochorial interface around GD12 followed later by a progressive rise in blood flow imposed by the rapid growth of the labyrinth towards term are important gestation stage-dependent environmental challenges determining oxygen exposure [32,33,34]. In this context, an important adaptive response to limit tissue damage involves a parallel temporal increase in placental antioxidant defenses as verified in normally progressing pregnancies [31,35,36]. Thus, the temporal changes in the expression of antioxidant enzymes SOD1, SOD2 and catalase, which are part of the first line of defense against oxidative damage, suggest that the placental ROS-detoxifying capacity in WKY pregnancies is sufficient to overcome tissue damage reflecting an effective adaptive response [36].

Two main deviations of the normal placental CSR profile were observed in SHRSP associated with progression of the SPE syndrome. First, both placental compartments exhibited a significant upregulation of CSR markers on GD14, indicating that onset of disease involves an exaggerate activation of the placental stress response and defective antioxidant expression. The presence of sustained oxidative stress due to persistent malperfusion would in turn accelerate the CSR resulting in premature ageing of the placenta. In particular, the failure to upregulate SOD2 expression towards GD18 observed in SHRSP placentas may be indicative of an underlying mitochondrial dysfunction, which is becoming increasingly recognized as a mechanism involved in preeclampsia pathogenesis [37,38,39,40]. Indeed, mitochondrial function appears to be differentially regulated in preeclampsia subtypes (i.e., early-onset vs. late-onset [38]) and mitochondrial adaptations including upregulated SOD2 activity play a paramount role in determining successful outcomes in preeclamptic pregnancies [39]. Of note, mitochondrial dysfunction can arise as a direct consequence of placental ischemia–reperfusion injury, as has been observed in the reduced uterine perfusion pressure (RUPP) model of preeclampsia [37] and also in healthy human placentas exposed to in vitro hypoxia/reoxygenation [39] It is therefore plausible that mitochondrial stress results from failed arterial remodeling in the SHRSP [11], which would allow highly pressured intermittent pulses to reach the placental site and cause ischemia–reperfusion damage to trophoblasts. Alternatively, a contribution of maternal pre-existing disease to mitochondrial dysfunction cannot be ruled out. Indeed, SHRSP display dysregulated basal expression of genes involved in mitochondrial ROS homeostasis (i.e., UCP2 [41], which in normal pregnancies has been shown to take part in the protective placental response associated with rapid growth of the labyrinth layer towards term [36]). Second, later on GD18, the sustained activation of stress pathways during earlier stages would compromise cell organizational processes involved in normal placentation, rendering the placenta with a limited ability to effectively adapt to later challenges. An earlier oxidative injury during the critical period of placental morphogenesis could account for the remarkable structural alterations observed in the SHRSP associated with placental insufficiency. For instance, AldhIA3-expressing glycogen trophoblasts, which are significantly depleted from the junctional zone in SHRSP placentas [11,42], appear to be selectively vulnerable to ischemia–reperfusion damage associated with enhanced endothelin 1 signaling [43]. Although the precise significance of these pathways in connection with SPE requires further investigation, our results suggest that placental insufficiency and consequent FGR in SHRSP could result from failed organizational processes during early placental morphogenesis in the context of sustained exaggerated activation of CSR pathways.

Hypertensive disorders of pregnancy impose a deviation of the placental glycopatterns and aberrant glycosylation of plasma proteins [44,45]. However, the relationship between the onset of disease and the placental glycophenotype has not been described. To the best of our knowledge, our study constitutes the first characterization of changes in galectin–glycan circuits associated with progression of SPE in chronic hypertensive pregnancies. Our lectin-based profiling of glycan expression revealed significant alterations in placental O-glycosylation coupled with maternal compartment-specific changes in sialylation associated with onset of disease in SHRSP pregnancies. Overall, SHRSP placentas displayed a significantly decreased expression of the TF antigen (core 1, Galβ1-3GalNAc) associated with onset of disease and downregulation of the Tn antigen (GalNAc) levels in the labyrinth layer as gestation progressed to GD18. Since core 1 O-glycans showed a reduction in SHRSP placentas, we hypothesize that progression of SPE is linked with an overall reduction of O-glycan synthesis. Another possible explanation is enhanced sialylation of the Tn antigen, which would interfere with HPA binding and thus prevent detection of the antigen. Since other glycan structures like core 1 and polyLacNAc sequences are reduced in SHRSP placentas but α2,6-sialylation seems to be increased, activity of the corresponding glycosyltransferase ST6GalNAc1 [46,47] might be responsible for increased sialylation of the Tn antigen because other structures are less available. Important pregnancy-relevant carriers of the sialyl-Tn antigen are mucin 1 (MUC1), CD44, integrins and osteopontin, all of which play roles in placental well-being [48,49,50,51]. In particular, loss of heavily O-glycosylated MUC1 in human placenta is associated with inflammation in pregnancies affected by preeclampsia and choriamnionitis [52,53]. MUC1 is also expressed in mouse trophoblast cells and its expression is regulated by peroxisome proliferator-activated receptor γ (PPAR-γ) [54], which is linked to preeclampsia in humans and rats [55,56,57].

While less prominent than for O-glycosylation, our study also revealed changes in *N*-glycans associated with SPE progression. Complex *N*-glycans belong to the major ligands for Gal-1 and 3 at cell surfaces [58]. The addition of GlcNAc in a β1,6 linkage to the *N*-glycan core by the enzyme Mgat5 initiates branching leading to tri- and tetra-antennary *N*-glycans, which display increased affinity for galectin binding [59]. Furthermore, Mgat5 products are the preferred substrate for addition of polyLacNAc sequences, which also enhances affinity for Gal-1 and -3 [60]. In this study, the plant lectins PHA-L and LEA were used to detect the Mgat5 product and polyLacNAc sequences, respectively. Of note, polyLacNAc sequences are also present on extended core 2 O-glycan structures. We revealed a reduction of Mgat5 products mainly in placental compartments of SHRSP implantation sites on GD14 (i.e., spongiotrophoblasts and giant cells), followed by decreased levels of polyLacNAc extensions later on GD18. These findings indicate that galectin binding may be impaired due to reduced amounts of their target sequences. Interestingly, galectin binding to polyLacNAc extensions of MUC1 O-glycans has been shown to prevent NK cell attack on tumor cells [61]. This finding suggests that reduced abundance of polyLacNAc sequences may indicate loss of MUC1 function and, hence, persistent inflammation in SHRSP placentas due to reduced protection against maternal immune cell activation.

Sialylation corresponds to the addition of sialic acids, which are typically found attached to the distal end of glycans. Sialic acids can be attached to O- as well as *N*-glycans via an α2,3 or α2,6 bond. Of note, the addition of α2,6-linked sialic acid prevents LacNAc recognition by Gal-1 and -9, possibly because these lectins primarily recognize terminal LacNAc units [60]. In contrast, Gal-3 is less influenced by this type of sialylation due to its ability to recognize internal LacNAc motifs [62]. In the current study, we observed an increase of both types of terminal sialylation in the maternal compartments of SHRSP implantations on GD14, followed by a decrease of α2,3-linked residues in the decidua and the labyrinth on GD18. The predominant increase of α2,3-sialylation on GD14 may be a compensatory mechanism to enhance galectin binding since, as discussed previously, other major glycan structures for galectin binding seemed to exhibit an overall reduction. Of note, important carriers of sialylated glycans in the placenta are PECAM-1 and VEGFR2 and it has been proposed that since endothelial cells express the enzyme for α2,6- as well as α2,3-sialyation, different sialyltransferases might be responsible for modulation of specific sialylation influencing homo- and heterophilic interactions and the binding of different angiogenic growth factors to these receptors [63,64]. In addition, it has been shown that sialylation is crucial for protection of fetal cells from maternal immune attack [65], suggesting that the reduction of α2,3-sialyation on GD18 in the decidua and labyrinth may account for increased maternal immune activation in these compartments.

A feature of preeclamptic placentas is dysregulation of the autophagy process, which prevents the maintenance of cellular homeostasis [66]. Interestingly, it has been shown that autophagy is important for human trophoblast functions, including invasion and vascular remodeling in extravillous trophoblasts, which are crucial for normal placental development [67]. Similar results have been observed in vivo by using a placenta-specific Atg7 (essential for autophagy) conditional knockout mouse model [68], which results in autophagy deficiency accompanied by poor placentation and elevated maternal blood pressure, suggesting its correlation with the pathophysiology of PE. Notably, both Gal-3 and Gal-9 have been implicated in the control of autophagy, acting as negative (Gal-3 [69]) and positive (Gal-9, mediated by polyLacNAc recognition [70,71]) modulators, respectively. These findings are in agreement with our observation that SHRSP implantations exhibited decreased Gal-9 and upregulated Gal-3 levels already at the onset of the disease and reduced polyLacNAc presentation (LEA staining) in placental compartments, which would possibly lead to impaired autophagy und thus cellular stress. Moreover, Sudhakar et al. showed that compromised autophagy due to loss of Gal-9 in mouse pancreatic and intestinal cells contributes to increased susceptibility to disease pathogenesis [71]. These findings led to the assumption that highly secretory cells (i.e., placental cells) are protected from endoplasmic reticulum stress and apoptosis via Gal-9-mediated autophagy that would otherwise lead to tissue inflammation or injury.

## 5. Conclusions

Overall, there seems to be a severe dysregulation of glycosylation in all compartments of SHRSP implantation sites, possibly caused by the early and sustained activation of CSR pathways (Figure 6). The observed changes in glycosylation which already occur on GD14 with the onset of disease before establishment of the maternal syndrome may account for altered cellular responses to environmental stimuli as well as cellular growth and differentiation patterns since glycans have multiple crucial roles in these processes. This assumption is further strengthened by the fact that galectins are important translators of the glycocode and the changes observed in this study for SHRSP implantations most likely led to reduced galectin binding on the cell surfaces. Thus, our findings uncovered a new role for the glycoredox status in promoting the development of SPE representing a promising mechanism in the pathogenesis of hypertensive disorders during gestation.

## Figures and Tables

**Figure 1 cells-10-00800-f001:**
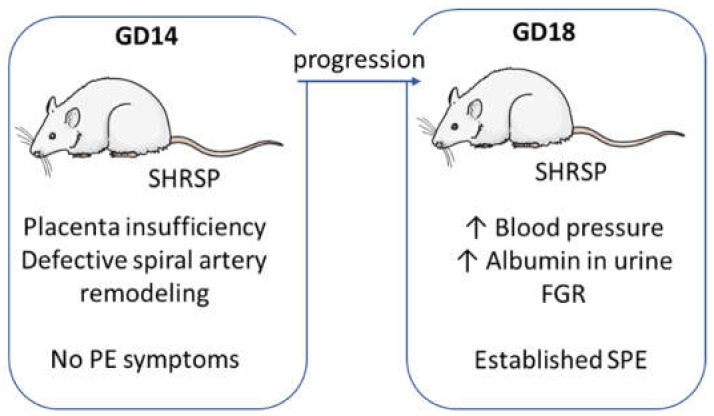
Summary of the superimposed preeclampsia (PE)-like features observed in SHRSP pregnancies. Gestation day (GD) 14 corresponds to the onset of disease, characterized by defective placentation and impaired arterial remodeling defects without full development of maternal features. As pregnancy progresses towards GD18, blood pressure rises steadily presenting the typical features observed in superimposed preeclampsia: aggravated hypertensive condition, end-organ damage (i.e., renal dysfunction with glomerular enlargement and proteinuria) and FGR (described in detail in [11]).

**Figure 2 cells-10-00800-f002:**
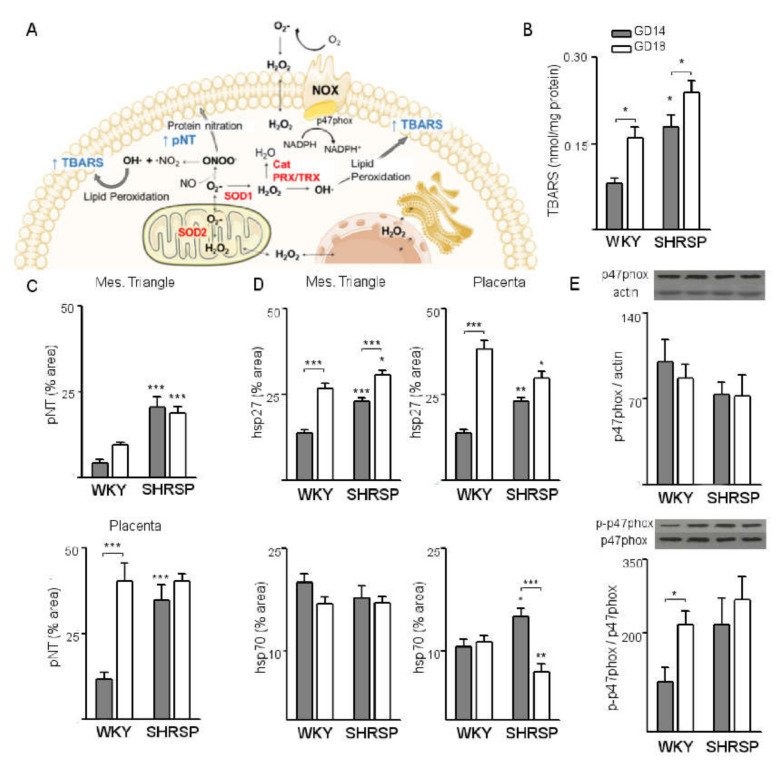
Sustained activation of placental CSR pathways in SHRSP pregnancies. (**A**) Schematic diagram summarizing the CSR and antioxidant defense pathways investigated in this study. NADPH oxidase (NOX) activation or increased activity of the mitochondrial electron flux leads to accumulation of superoxide (O_2_•−), which under normal conditions is rapidly dismutated to hydrogen peroxide (H_2_O_2_) by superoxide dismutases (SOD1, SOD2). H_2_O_2_ can readily diffuse through membranes to react with proteins and DNA and is detoxified by cellular peroxidases (catalase (Cat) and the peroxiredoxin/thioredoxin systems (PRX/TRX)). Failed detoxification of ROS produces peroxynitrite (ONOO−) and hydroxyl radicals (OH•), promoting the nitration of membrane proteins and the lipid peroxidation cascade. (**B**) Analysis of lipid peroxidation in both pregnancy models as assessed by quantification of placental TBARS. (**C**) Analysis of protein nitrotyrosine (pNT) levels in maternal (mesometrial triangle, top panel) and trophoblastic (placenta, bottom panel) layers of WKY and SHRSP implantations as analyzed by immunohistochemistry. (**D**) Immunohistochemical analyses of the expression of CSR chaperones hsp27 (upper panels) and hsp70 (bottom panels) in both models. (**E**) Evaluation of placental NOX activation in both pregnancy models as indicated by Western blot analyses of total and phosphorylated p47phox levels. The results are expressed as the means ± SEM, *n* = 5–7 animals/group. In all the panels, * *p* < 0.05, ** *p* < 0.01 and *** *p* < 0.001 as evaluated by two-way ANOVA and Bonferroni post-hoc tests.

**Figure 3 cells-10-00800-f003:**
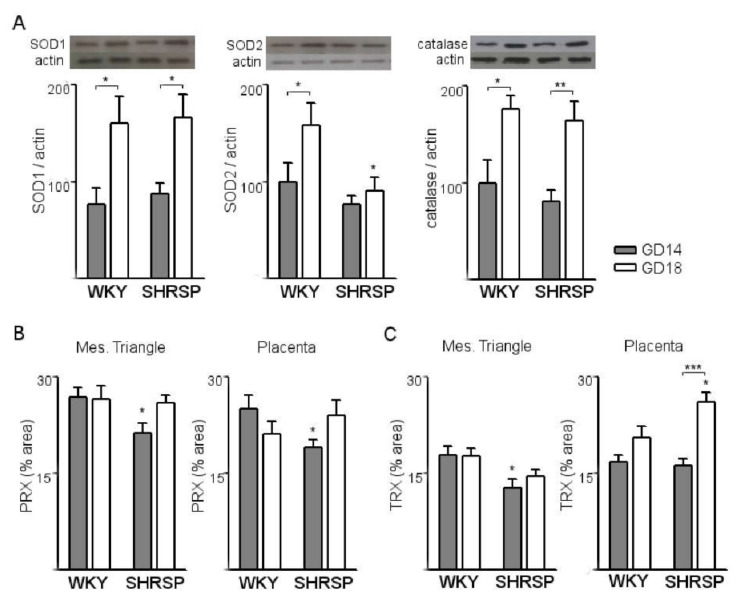
SHRSP placentas display defective expression of antioxidant systems. (**A**) Western blot analyses of the expression of ROS-detoxifying enzymes SOD1 (left panel), SOD2 (middle panel) and catalase (right panel) in WKY vs. SHRSP placentas. Normal upregulation of SOD2 expression towards GD18 was abrogated in SHRSP pregnancies. (**B**) Immunohistochemical analysis of PRX expression in both models. (**C**) Placental expression levels of TRX in both models as assessed by immunohistochemistry. The results are expressed as the means ± SEM, *n* = 5–7 animals/group. In all the panels, * *p* < 0.05, ** *p* < 0.01 and *** *p* < 0.001 as determined by two-way ANOVA and Bonferroni post-hoc tests.

**Figure 4 cells-10-00800-f004:**
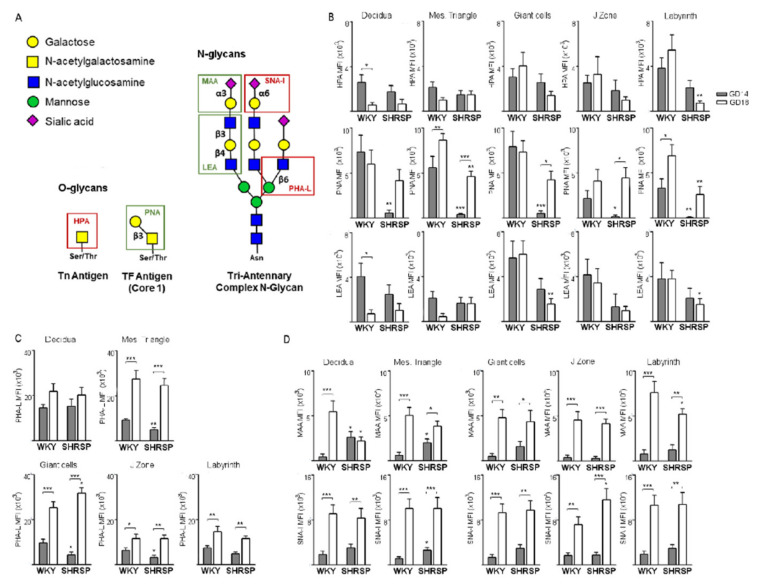
Lectin-based profiling of the placental glycophenotype associated with SPE progression. (**A**) Schematic representation of the *O*- and *N*-glycan moieties recognized by the lectin panels used in the present study. (**B**) Placental expression levels of O- (HPA and PNA lectins, upper and middle panels, respectively) and polyLacNAc-extended glycans (LEA, bottom panels) in both models. Results are expressed as mean fluorescence intensity (MFI). (**C**) Analysis of the expression of complex branched *N*-glycans in the different placental layers as assessed by PHA-L binding. (**D**) Levels of terminal α2,3- (MAA, top panel) and α2,6-sialylation (SNA-I, bottom panel) in both pregnancy models. The results are presented as the means ± SEM, *n* = 5–7 animals/group. In all the panels, * *p* < 0.05, ** *p* < 0.01 and *** *p* < 0.001 as evaluated by two-way ANOVA and Bonferroni post-hoc tests.

**Figure 5 cells-10-00800-f005:**
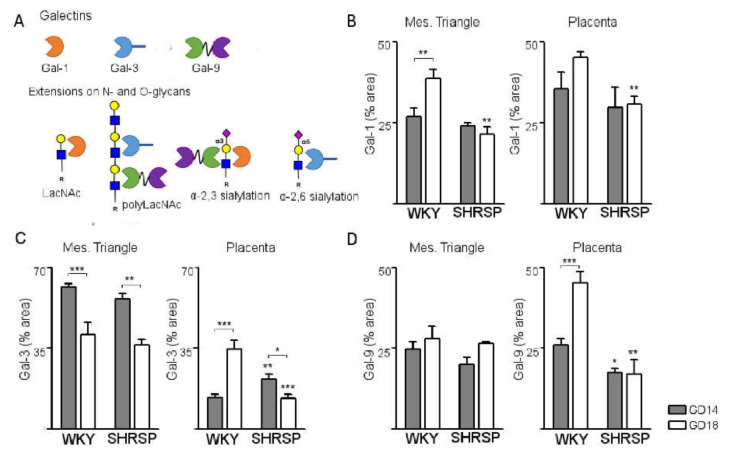
Temporal variation of the placental galectin signature associated with the development of SPE in SHRSP pregnancies. (**A**) Schematic diagram depicting the specific glycan moieties recognized by the different stress-sensitive galectins determined in this study. (**B**) Immunohistochemical analysis of Gal-1 expression in maternal (mesometrial triangle, left panel) and trophoblastic layers (placenta, right panel) of WKY and SHRSP implantations. SHRSP placentas failed to upregulate Gal-1 expression as gestation progressed to GD18. (**C**) Placental galectin-3 expression levels in both pregnancy models. Onset of SPE was marked by a significant upregulation of Gal-3 in trophoblastic compartments of the SHRSP placenta. (**D**) Immunohistochemical evaluation of Gal-9 expression in both models. SHRSP displayed decreased trophoblastic expression of Gal-9 on GD14, with a failure to upregulate levels of this lectin towards GD18. The results are presented as the means ± SEM, *n* = 5–7 animals/group. In all the panels, * *p* < 0.05, ** *p* < 0.01 and *** *p* < 0.001 as evaluated by two-way ANOVA and Bonferroni post-hoc tests.

**Figure 6 cells-10-00800-f006:**
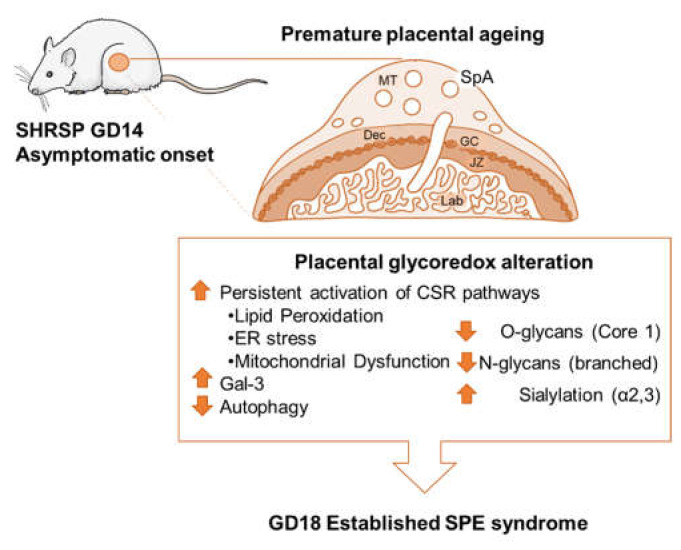
Summary of the placental molecular pathways involved in the pathogenesis of SPE in SHRSP pregnancies. Abbreviations: MT, mesometrial triangle; GC, giant cells; JZ, junctional zone; SpA, spiral artery. We hypothesize that as a result of placental malperfusion, early and sustained activation of CSR pathways intimately linked to impaired glycosylation (i.e., dysregulation of the so-called glycoredox system) would cause a premature aging of the placenta. This response would in turn provoke exhaustion of the placental defense systems, rendering the placenta unable to effectively adapt to later challenges. This local stress response is finally translated systemically leading to full establishment of the SPE syndrome towards term.

## Data Availability

All data presented in this study are contained within the article or its supplementary information.

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
