# Peer review of "Placental Glycoredox Dysregulation Associated with Disease Progression in an Animal Model of Superimposed Preeclampsia"

_cells, 2021, doi:10.3390/cells10040800_

Round 1

Reviewer 1 Report

General comments:

The study presented by Blois et al. has a great merit of investigating the mechanistic regulation of the superimposed preeclampsia in rats. The focus made on comparative expression pattern of cellular stress markers and their influence on galectin-glycan circuits during disease development is valuable. The finding of the authors concerning a new role of the glycoredox balance in the pathogenesis of superimposed preeclampsia not only contributes to improve our understanding of the pathogenesis of the disease but offers additional valuable knowledge for the design of new therapeutic/prophylactic approaches which might efficiently help against the mentioned disease. 

The paper submitted for evaluation and acceptance for the publication is very well structured and intelligibly written. The experimental models chosen by the authors are appropriate, and all methods employed are very well described, allowing a reproduction of the study by others. The results are accurately and convincingly described. The same is true for the discussion of their finding. And, conclusions drawn are supported by the presented data.  

Nonetheless, there are still few mistakes that must be corrected prior to the final acceptance of the proposed study for publication. Additionally, this reviewer also recommends the authors to avoid using extremely long sentences (more than 5 lines), which unfortunately relatively frequent in the discussion section. Such long sentences could be bothering for the readers. Therefore, when applicable, more concise sentences must be privileged.

Each of these too long sentences should probably be spilt into several sentences.

Specific comments

Abstract: No comment

Introduction: No comment

Materials and Methods:

Line 113, please add comma after the word “water”

Line 140, please check, do you really mean 2020 min?

Line 167, please change to “in PBS” instead of “inPBS”

Discussion:

Line 403 to 408: this sentence appears too long. Please, split it into several sentences.

Line 423 to 428: this sentence appears too long. Please, split it into several sentences.

 Line 441 to 447: this sentence seems too long. Please, split it into several sentences.

Line 447 to 452: this sentence appears too long. Please, split it into several sentences.

Reviewer 2 Report

The current study by Blois and co-workers addresses the role of placental glycoredox status regarding the progression of preeclamsia.  The study is very well designed and written.  One minnor comment in response to the data shown in Figure 3. SOD2 is a mitochondrial protein, thus, the normalizer should be mitochondrial as well. 

Reviewer 3 Report

The paper “Placental Glycoredox Dysregulation Associated with Disease Progression in an Animal Model of Superimposed Preeclampsia” by Blois et al. describes that the balance of oxidant and anti-oxidant is important for the pathogenesis of superimposed preeclampsia. And autophagy is involved in the process. In general, the purpose of this study is attractive. However, the some problems lessen the overall quality of the paper. All concerns listed below and the part of discussion should be easy-to-grasp for improving the overall quality.

Major concerns

P2, L89-102: Such this kind of content is usually not necessary in the part of Materials and methods; a line or two is sufficient.

P3, L118, 183: Since paragraphs 2.2 and 2.7 are both in materials and methods related to immunohistochemistry, they should be combined into one.

P7, L279-284: “Furthermore, western blot analyses revealed that normal pregnancy progression was associated with a significant upregulation of phosphorylated p47 phox (Figure 2E) from GD14 to GD18, indicating increased NADPH oxidase 2 (NOX2) activation. Of note, SHRSP implantations displayed increased levels of phospho-p47 phox earlier on GD14, which remained similarly elevated as pregnancy progressed to GD18.” Is there a statistical significance?

P9, L368-P10, L378: Why are there such differences in the changes of Gal along the course of pregnancy between SHR and WKY?

P11, L444-446: “ischemia reperfusion injury to trophoblasts, as has been observed in the RUPP model of preeclampsia in which hypertension is a direct consequence of uteroplacental ischemia” In this animal study [37], ischemia-reperfusion phenomenon impaired placentation, but that was because it was performed to confirm the effect of ischemia-reperfusion on the placentation, and I don't think that phenomenon is also accepted to human. Is there any evidence for that?

Figure 6: It's a well-drawn diagram; however, can't they make the correlation diagram more understandable to everyone?

Is it possible to compare serum concentrations of oxidants or anti-oxidants with placental concentrations?

Round 2

Reviewer 3 Report

The authors replied on each comment sincerely and the replies were appropriate. The quality of papers submitted for consideration includes enough reader's interest and scientific quality. The given paper satisfies requirements for publication of this journal.